# *Vibrio cholerae* RbmB is an α-1,4-polysaccharide lyase with biofilm-disrupting activity against Vibrio polysaccharide (VPS)

Ranjuna Weerasekera[1], Alexis Moreau[2], Xin Huang[2,3], Kee-Myoung Nam[2], Alexander J. Hinbest[1], Yun Huynh[1], Xinyu Liu[4], Christopher Ashwood[5,6], Lauren E. Pepi[5,6], Eric Paulson[3], Lynette Cegelski[4], Jing Yan[2,7] *, Rich Olson[1] *

1 Department of Molecular Biology and Biochemistry, Molecular Biophysics Program, Wesleyan University, Middletown, Connecticut, United States of America, 2 Department of Molecular, Cellular, and Developmental Biology, Yale University, New Haven, Connecticut, United States of America, 3 Department of Chemistry, Yale University, New Haven, Connecticut, United States of America, 4 Department of Chemistry, Stanford University, Stanford, California, United States of America, 5 Glycomics Core, Beth Israel Deaconess Medical Center, Harvard Medical School, Boston, Massachusetts, United States of America, 6 Department of Surgery, Beth Israel Deaconess Medical Center, Harvard Medical School, Boston, Massachusetts, United States of America, 7 Quantitative Biology Institute, Yale University, New Haven, Connecticut, United States of America

* jing.yan@yale.edu (JY); rolson@wesleyan.edu (RO)

**Data Availability Statement:** The simulation code has been uploaded to a public GitHub repository, available here: https://github.com/kmnam/RbmB-

## Abstract

Many pathogenic bacteria form biofilms as a protective measure against environmental and host hazards. The underlying structure of the biofilm matrix consists of secreted macromolecules, often including exopolysaccharides. To escape the biofilm, bacteria may produce a number of matrix-degrading enzymes, including glycosidic enzymes that digest exopolysaccharide scaffolds. The human pathogen *Vibrio cholerae* assembles and secretes an exopolysaccharide called VPS (Vibrio polysaccharide) which is essential in most cases for the formation of biofilms and consists of a repeating tetrasaccharide unit. Previous studies have indicated that a secreted glycosidase called RbmB is involved in *V. cholerae* biofilm dispersal, although the mechanism by which this occurs is not understood. To approach the question of RbmB function, we recombinantly expressed and purified RbmB and tested its activity against purified VPS. Using a fluorescence-based biochemical assay, we show that RbmB specifically cleaves VPS *in vitro* under physiological conditions. Analysis of the cleavage process using mass spectrometry, solid-state NMR, and solution NMR indicates that RbmB cleaves VPS at a specific site (at the α-1,4 linkage between D-galactose and a modified L-gulose) into a mixture of tetramers and octamers. We demonstrate that the product of the cleavage contains a double bond in the modified guluronic acid ring, strongly suggesting that RbmB is cleaving using a glycoside lyase mechanism. Finally, we show that recombinant RbmB from *V. cholerae* and the related aquatic species *Vibrio coralliilyticus* are both able to disrupt living *V. cholerae* biofilms. Our results support the role of RbmB as a polysaccharide lyase involved in biofilm dispersal, as well as an additional glycolytic enzyme to add to the toolbox of potential therapeutic antibacterial enzymes.

lyase-simulations.git The code is one script, named simulate_RbmB.py, which upon running generates a set of .csv data files and (a version of) the figure used in the paper.

**Funding:** Research reported in this publication was supported by the National Institute of General Medical Sciences of the National Institutes of Health (https://www.nigms.nih.gov/) under award number R15GM152959 to RO and 1DP2GM146253 to JY. JY also acknowledges support from the National Science Foundation, Division of Materials Research (https://www.nsf.gov/div/index.jsp?div=DMR) under award number 2205006 and the Alfred P. Sloan Foundation under research fellowship FG-2023-20857. LC acknowledges support from the National Institutes of Health under award number R01GM117278. The funders did not play any role in the study design, data collection and analysis, decision to publish, or preparation of the manuscript.

**Competing interests:** The authors declare no competing interests.

## Author summary

Biofilms are protective sheaths produced by many bacteria that play important roles in environmental survival and in hosts during infection. Understanding how biofilms form and disperse is essential in the fight against harmful bacterial pathogens. *Vibrio cholerae* is a historically significant human pathogen that forms biofilms made of a polysaccharide called VPS and secreted accessory proteins. Within the *V. cholerae* biofilm gene cluster is a gene encoding a protein called RbmB, which is known to play a role in biofilm dispersal. Using a combination of biochemical and analytical tools, we show that RbmB indeed cleaves VPS into tetra- and octasaccharide fragments at a unique site and demonstrate that it falls into the family of polysaccharide lyase enzymes (as opposed to the more common hydrolases). Furthermore, recombinant RbmB disrupts living *V. cholerae* biofilms through VPS digestion, further indicating that it may play a role in VPS processing or degradation.

## Introduction

The formation of biofilms is a common survival strategy of bacteria. Biofilms consist of a community of cells encapsulated in a matrix of secreted biomolecules. The composition of the matrix varies between different bacterial species, but may contain polysaccharides, proteins, and nucleic acids [1,2]. Following biofilm establishment, cells may eventually seek to escape the matrix, typically in response to environmental signals or nutrient status [3,4]. At this time, enzymatic factors are implemented in digesting components of the matrix, leading to biofilm disruption and cell dispersal. For example, the glycoside hydrolase dispersin B, produced by the periodontal biofilm-forming pathogen *Aggregatibacter actinomycetemcomitans*, specifically cleaves the polysaccharide poly-*N*-acetylglucosamine (PNAG) by hydrolyzing β-1,6 glycosidic linkages [5] leading to biofilm disruption [6].

The human pathogen *Vibrio cholerae* produces biofilms in aquatic environments controlled through a complex regulatory mechanism involving cell-cell signaling [7,8]. The transition from a planktonic to a biofilm lifestyle is also associated with hyperinfectivity phenotypes [9,10] and ingestion of biofilm-embedded bacteria leads to higher incidence of disease in human populations [11]. The biofilm-forming system is encoded within a cluster of ~25 genes, of which the majority are responsible for producing the machinery that assembles and secretes an exopolysaccharide called Vibrio polysaccharide (VPS) [12,13]. Production of VPS is necessary for *V. cholerae* biofilm formation in most cases [12] and mutations causing defective VPS biogenesis lead to an inability to form biofilms [13]. In addition to VPS, *V. cholerae* secretes several matrix proteins, including RbmA, RbmC, and Bap1, which are involved in interactions between cells and VPS [14], assembling the matrix [15,16], adhesion to various biotic and abiotic surfaces [17,18], and biofilm hydrophobicity and elasticity [19]. Finally, the *V. cholerae rbm* gene cluster encodes a protein, RbmB, with predicted sequence and structural similarity to polysaccharide glycosidases [15]. Deletion of *rbmB* is associated with excess VPS accumulation, which is consistent with a role for RbmB as a VPS-digesting enzyme [15] that could play a role in *V. cholerae* biofilm dispersal (Fig 1A). Furthermore, genetic screening identified *rbmB* as one of several potential genes that influence *V. cholerae* biofilm dispersal, as quantified through changes in plastic-associated biomass after 16 hours [20]. These major phenotype associations implicate RbmB as a VPS glycosidase involved in biofilm dispersal. Yet, the biochemistry that underlies *V. cholerae* biofilm dispersal is currently not well understood and no direct enzymatic activity of RbmB has been demonstrated.

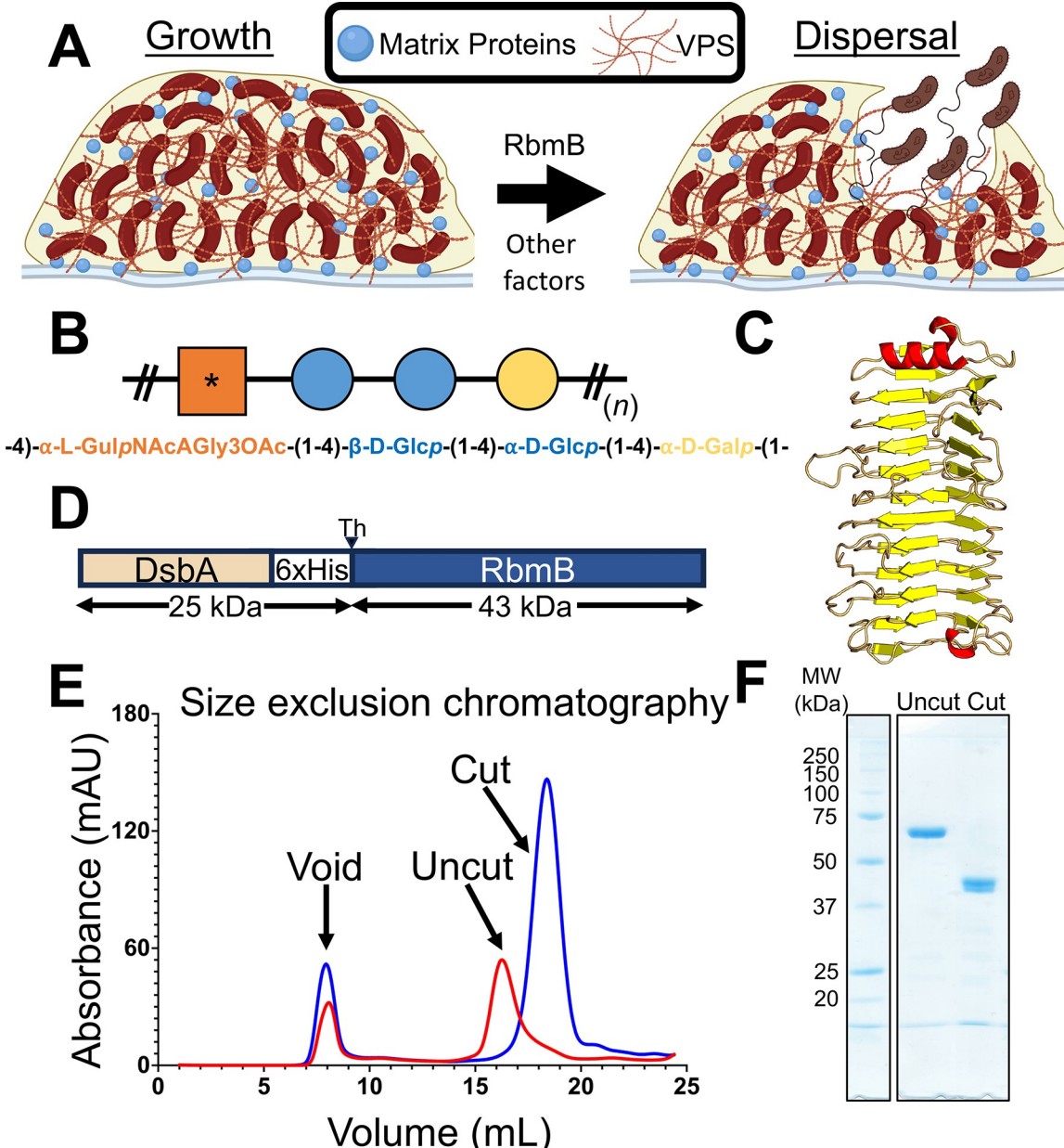

**Fig 1. RbmB plays a role in *V. cholerae* biofilm dispersal.** (A) Schematic of a working model suggesting cellular escape is helped by RbmB-mediated cleavage of VPS. Created in BioRender. Olson, R. (2022) BioRender.com/m42l648 and Olson, R. (2022) BioRender.com/n45c416. (B) Schematic version of the tetrasaccharide structure of VPS based on [21]. The asterisk represents the modified gulose moiety. (C) AlphaFold2 [24] model for RbmB structure. (D) RbmB was expressed as a thrombin-cleavable fusion protein with DsbA and a periplasmic secretion signal peptide at the N-terminus. (E) Size exclusion chromatography trace showing purified RbmB fusion before and after cleavage of the DsbA fusion partner. (F) SDS-PAGE gel showing purified DsbA-RbmB fusion and cleaved and purified RbmB. The expected molecular weight for cleaved RbmB is ~42.7 kDa.

 The chemical structure of VPS is reported [21] to consist of a repeating tetrasaccharide containing a uniquely modified L-gulose moiety (N-acetylated on C2, partially O-acetylated on C3, and containing an amide-linked glycine on C6) followed by two glucose and one galactose monosaccharides (Fig 1B). The linkages between the monosaccharides are all α-1,4, except for a β-1,4 glycosidic bond between the two glucose moieties. This unusual structure raises

questions about polysaccharide specificity for the VPS-interacting matrix proteins as well as for the glycosidic activity of RbmB.

Glycosidic enzymes are generally categorized according to their general mechanism of action as either hydrolases, which use water as a nucleophile to catalyze the breakage of the glycosidic bond, or lyases, which employ mechanisms other than hydrolysis and often result in the formation of a double bond in the ring structure [22]. More specifically, polysaccharide lyases break the glycosidic bond through an E1cB elimination reaction utilizing the uronic acid group [22], introducing a double bond between C4 and C5 in the sugar ring following cleavage of the glycosidic bond. While RbmB is annotated as a family 28 glycosyl hydrolase in the NCBI sequence database (via a BLAST search [23]), the presence of the carbonyl group on C6 of the gulose sugar in VPS raises the possibility that RbmB could function as a lyase through cleavage of the galactose-gulose glycosidic bond. This is further supported by analysis of a three-dimensional structure prediction by AlphaFold2 [24], which predicts a parallel β-helical arrangement similar to pectate/pectin lyases (Fig 1C) [25]. Furthermore, a variety of protein family prediction algorithms including InterPro [26], CATH [27], and SMART [28] also place RbmB within the pectin lyase family.

In order to investigate the putative glycosidase activity of RbmB against VPS, we produced recombinant RbmB through heterologous expression in *Escherichia coli* and conducted multiple assays to evaluate the ability of RbmB to digest isolated VPS *in vitro*. Using a combination of liquid chromatography, mass spectrometry, solid-state nuclear magnetic resonance (NMR), and solution NMR, we demonstrate that RbmB is indeed a polysaccharide lyase which cleaves the α-1,4 glycosidic bond between the galactose and gulose units in VPS. Our results are consistent with a model wherein RbmB recognizes a footprint longer than eight monosaccharides on VPS and endolytically cleaves the polymer into a combination of tetra- and octasaccharides. Finally, we show that recombinantly produced RbmB can efficiently disrupt *V. cholerae* biofilms, adding direct support for the ability of RbmB to influence *V. cholerae* biofilm dynamics.

## Results

### Heterologous expression and purification of RbmB

An expression pipeline for producing RbmB was created to permit VPS digestion assays and biofilm experiments. Initial attempts to express RbmB in *E. coli* alone or as a fusion protein yielded insoluble or inactive protein. Noting that RbmB contains four cysteine residues (two disulfides in the predicted structure), we reasoned that inadequate formation of disulfide bonds in the cytoplasm might lead to inactive protein. Thus, RbmB was cloned into pET39b (Novagen) to create a thrombin-cleavable N-terminal DsbA fusion (a protein that catalyzes the formation of disulfide bonds) containing a native periplasmic signal peptide (Fig 1D). Expression and purification of this construct in *E. coli* yielded soluble protein that ran as a monodisperse peak on a size-exclusion column (Fig 1E). Following cleavage of the DsbA fusion partner from RbmB, RbmB maintained solubility and monodispersity (Fig 1E). The purified RbmB fusion and cleaved RbmB ran at their expected molecular weights on an SDS-PAGE gel (Fig 1F).

### Recombinant-purified RbmB disrupts Bap1/VPS aggregates

To assess whether recombinant RbmB displays glycolytic activity against VPS, we used a previously described assay [18] making use of a GFP$_{UV}$-tagged deletion mutant of the *V. cholerae* adhesin Bap1 (with a deletion of 57 amino acids, which removes an insoluble loop required for adhesion——for simplicity we refer to this truncated protein as Bap1 throughout the rest of this paper). We previously observed that incubation of Bap1 with VPS leads to aggregates too

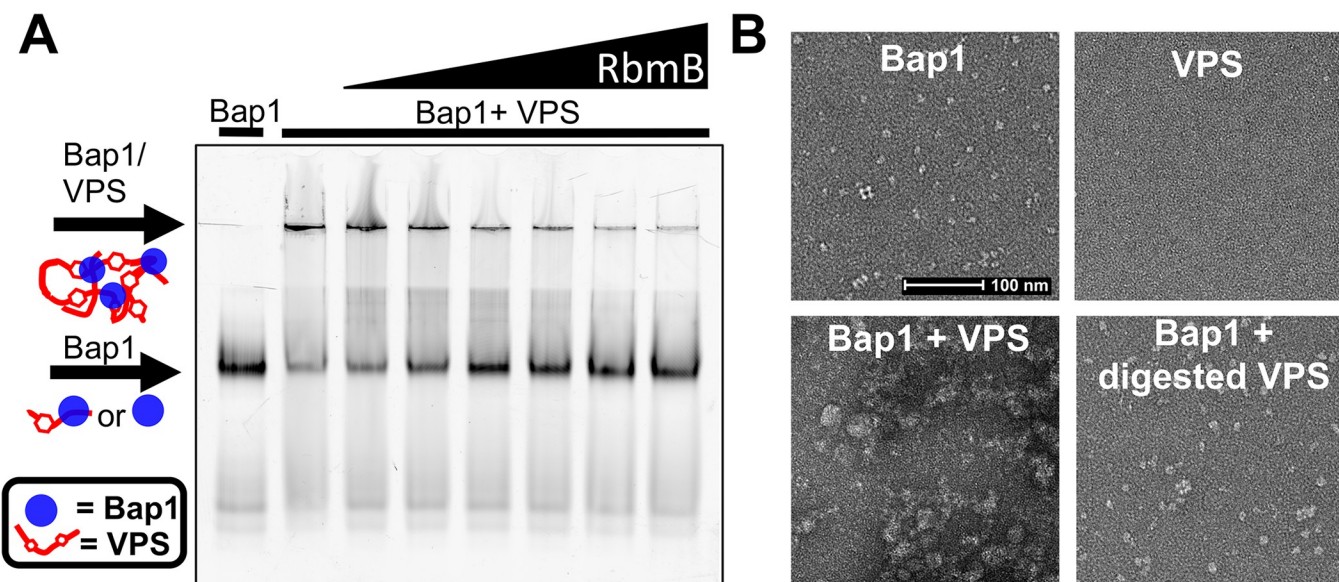

**Fig 2. RbmB digests VPS *in vitro*.** (A) Fluorescent-imaged native-PAGE gel using Bap1-GFP$_{UV}$ to label VPS polymer. Addition of Bap1 to uncut VPS shows a shift to a large complex that is unable to enter the gel. Increasing amounts of RbmB show a decrease in the aggregates and increase in free Bap1, suggesting cleavage of VPS into small fragments. Lanes left to right: Bap1 alone, Bap1+VPS, Bap1+VPS pre-treated with 0.05 μg RbmB, Bap1+VPS pre-treated with 1 μg RbmB, Bap1+VPS pre-treated with 2.5 μg RbmB, Bap1+VPS pre-treated with 5 μg RbmB, Bap1+VPS pre-treated with 10 μg RbmB, and Bap1+VPS pre-treated with 15 μg RbmB. (B) Negative-stained EM images of Bap1/VPS complexes. Bap1 particles form dense aggregates of size >0.5 μm when added to VPS. Digestion with RbmB disrupts aggregates back to smaller Bap1/VPS particles of size <0.1 μm.

large to migrate into the 5% acrylamide stacking gel using native gel electrophoresis (Fig 2A). We hypothesized that cleavage of VPS into smaller fragments might prevent aggregate formation after incubation with Bap1. To test this hypothesis, we incubated 5 μg of VPS with increasing amounts of recombinant DsbA-RbmB (0, 0.5, 1, 2.5, 5, 10, and 15 μg), stopped the reaction by boiling for 5 minutes, and added 5 μg of Bap1-GFP$_{UV}$. The entire sample was then run on a 10% acrylamide native gel with a 5% stacking gel. Indeed, we observed an RbmB-concentration-dependent decrease in the amount of Bap1/VPS aggregate and a concomitant increase in the amount of monomeric Bap1 detected in the gel (Fig 2A and quantification in S1 Fig). We interpret this result as cleavage of VPS into smaller fragments unable to shift the apparent electrophoretic mobility of Bap1.

As a second measure of RbmB's action on the VPS polymer, we employed negative-stain electron microscopy (EM) to visualize the formation of Bap1/VPS aggregates. Bap1 monomers are clearly visible by negative-stain EM, whereas purified VPS polymer could not be visualized due to a lack of sufficient molecular density and contrast. However, addition of Bap1 to VPS yielded large observable aggregates, consistent with the behavior observed on the native acrylamide gels (Fig 2B). Treatment of VPS with RbmB overnight followed by incubation with Bap1 yielded images more closely resembling Bap1 alone; we did not observe any of the large aggregates associated with untreated VPS and Bap1 (Fig 2B), although some small remnant aggregates were detected. This qualitative result is consistent with the gel-shift assay results in that treatment with RbmB appears to prevent the formation of large aggregates by Bap1, possibly by cutting VPS into smaller fragments.

## Biochemical characterization of VPS cleavage by RbmB

To further assess whether RbmB cleaves glycosidic bonds in VPS, we used an assay to quantify the amount of reducing ends produced following incubation with RbmB. This assay involves

heating VPS in the presence of boric acid and ethanolamine, forming a fluorescent molecule that can be monitored (excitation peak at 357 nm and emission peak at 443 nm) and quantified using a standard curve of a reducing-sugar monosaccharide (S2 Fig). It is expected that each glycosidic bond cleavage event by RbmB should produce one new reducing end, allowing us to monitor the pace of cleavage over time.

Using this assay, we observed that treatment of 25 μg of VPS with 20 μg of RbmB at 30˚C yielded an increase in the concentration of reducing ends over the span of several hours, with the reaction reaching a plateau after approximately 2 hours (Fig 3A). As a negative control, a VPS sample to which RbmB was not added did not show any increase in signal over time. Thus, treatment with RbmB causes the formation of new reducing ends, consistent with cleavage of glycosidic linkages. Using this assay, we were also able to assess the dependence of this cleavage on several environmental variables, including pH, temperature, and ionic strength. RbmB displays a maximal increase in reducing ends with pH values between pH 6 and 9 (Fig 3B), an optimal temperature range of 30–40˚C (Fig 3C), and an optimal ionic strength below ~350 mM NaCl (Fig 3D). We also tested the dependence of the reaction on various ions and observed an approximate two-fold increase in activity in the presence of calcium (Fig 3E) over protein treated with EDTA or EGTA. Calcium could assist in the cleavage mechanism of RbmB, as has been observed with some polysaccharide lyases [29]. All of these results are consistent with enzyme-dependent cleavage of VPS under physiological conditions likely encountered by *V. cholerae* in the environment and in host organisms.

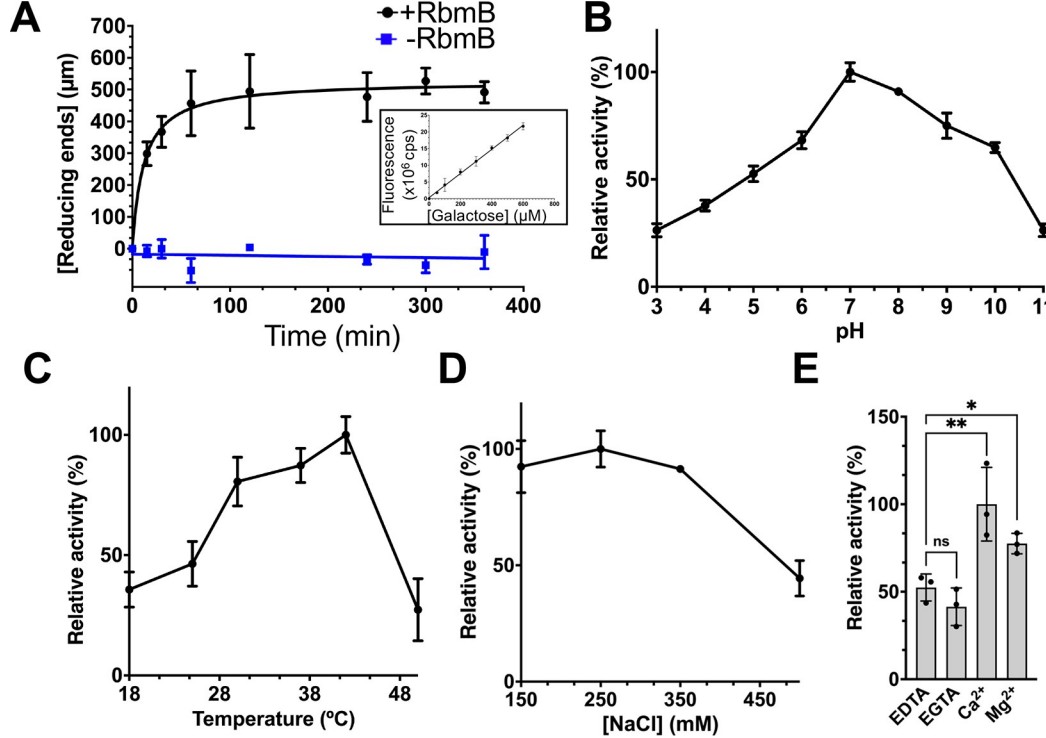

**Fig 3. VPS cleavage activity of purified RbmB.** (A) Quantification of reducing ends as a function of time formed during RbmB-mediated cleavage of VPS. A standard curve using galactose as a substrate is shown in the inset and S2 Fig. The effects of pH (B), temperature (C), salt concentration (D), and divalent cations (E) on the activity of RbmB against VPS are also shown. All data are represented as the mean ± SD (*n* = 3). 100% activity in each graph is calculated based on the condition with the highest activity. Statistical significance was determined using an unpaired, two-tailed *t*-test with Welch's correction. ns = not significant, *p<0.05, **p<0.01. Solid lines indicate which samples were compared in the statistical tests. All experiments were conducted with 25 μg of VPS and 20 μg of RbmB.

To gain insight into the specificity of RbmB for VPS, we tested the activity of the enzyme using the same reducing end assay against a variety of different carbohydrate polymers, including sodium alginate from marine algae (heterogeneous mixture of 1,4-linked β-D-mannuronic acid and α-L-guluronic acid), hydroxyethyl cellulose (HEC, β-1,4-linked modified D-glucose units), arabinan from sugar beet (α-1,5-linked arabinose with some α-1,3- and α-1,2-linked L-arabinofuranosyl branches), and β-D-glucan from yeast (β-1,3-linked D-glucose). RbmB did not appear to digest any of the non-VPS polymers (S3 Fig). Interestingly, we also saw some activity of alginate lyase against VPS. This may arise from some structural similarities between alginate and VPS, notably the presence of an α-L-guluronic acid core in each polysaccharide (and the fact that α-L-guluronic acid is a C5 epimer of D-mannuronic acid, also present in alginate). As a positive control, we tested the enzyme alginate lyase (poly-β-D-mannurate lyase, Sigma) against the same sodium alginate sample and saw robust activity. These results suggest that RbmB is not a generic glycosidase, although we only tested a few polymer substrates and a more exhaustive search, particularly against exopolysaccharides from other biofilm-forming bacteria, will be necessary to outline how promiscuous RbmB is against a wider range of natural polysaccharides.

## RbmB cleaves VPS at a single position yielding a mixture of tetra- and octasaccharides

To further understand the nature of VPS cleavage by RbmB, we utilized liquid chromatography–mass spectrometry (LC/MS) to analyze the products of RbmB cleavage of VPS. An overnight cleavage reaction of VPS by RbmB was carried out, the sample was purified using solid-phase extraction (SPE), and the products were run over a porous graphitized carbon (PGC) column coupled to a mass spectrometer run in negative mode (Orbitrap Fusion Lumos Tribrid Mass Spectrometer, Thermo Fisher Scientific). Separation by liquid chromatography yielded two primary peaks (Fig 4A), with masses consistent with tetrameric (Fig 4B) and octameric (Fig 4C) VPS species based on the reported structure of VPS in the literature [21]. This result supported the notion that RbmB may be cleaving VPS at only one of the four possible (and unique) glycosidic bonds. Otherwise, we would expect to observe a mixture of mono-, di-, and trisaccharides.

If RbmB were a hydrolase, the predicted mass from cleaving at any one of the four glycosidic bonds would likely result in fragments of the same mass as the VPS tetrasaccharide building block. However, if RbmB were a lyase cleaving before the gulose residue, the resulting elimination reaction might lead to a product missing a hydroxyl group on C4 of the gulose and the emergence of a double bond between C4 and C5. The mass determined for the tetrameric species (759.23 Da) is consistent with this outcome, suggesting that RbmB may be acting as a lyase and cleaving the α-1,4 glycosidic bond between the galactose and gulose moieties (Fig 4D). If RbmB were acting as a hydrolase, we would expect a molecular weight of 776.24 Da for the tetramer, which we do not observe in the mass spectrum. The published characterization of VPS indicated that 20% of the second glucose moieties were further modified with an N-acetyl group and that an additional O-acetyl group was found on C3 of the L-gulose sugar [21]. In our spectra, we see peaks corresponding to masses consistent with VPS without either modification, VPS with an O-acetyl group alone, and VPS with both O-acetyl and N-acetyl groups (Fig 4D and S1 Table).

To further confirm that RbmB cleaves the glycosidic bond between the galactose and L-gulose units, we performed an additional mass spectrometry experiment including an MS2 fragmentation step. As the three unmodified hexose sugars have identical masses, we treated RbmB-cleaved VPS with NaBH$_4$ to modify the reducing end (leading to an increase in the

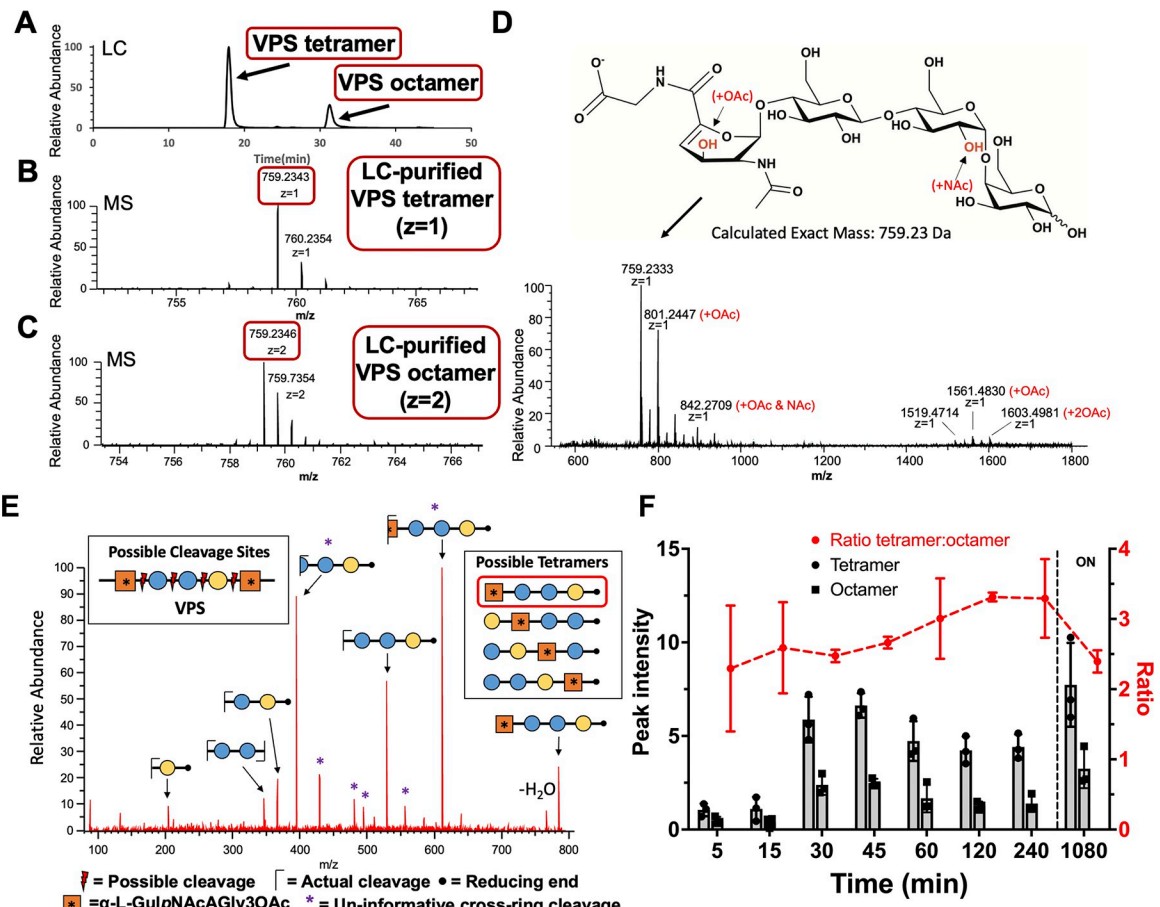

**Fig 4. Mass spectrometry analysis of VPS fragments produced by RbmB cleavage.** (A) Liquid chromatographic analysis of an overnight digestion of VPS by RbmB indicates a separable mixture of tetra- and octameric oligosaccharides of VPS. (B,C) Mass spectrometry analysis of liquid chromatography peaks indicates a primary species consistent with the published tetrameric form of VPS, including formation of a C4-C5 double bond in the gulose moiety due to a lyase mechanism. (D) Full negative mode mass spectrometry spectrum showing modified species and the chemical structure of the primary species. The spectra also indicate species consistent with O-acetylation and N-acetylation of the tetrasaccharide, presumably on the gulose and glucose saccharides as indicated in red. Annotation of major peaks in the spectrum is shown in S1 Table. (E) MS2 analysis is consistent with cleavage between the galactose and gulose moieties. Four possible cleavage sites would produce four different tetrameric species (shown in inset boxes). To identify the exact cleavage site, VPS was borate-reduced following RbmB cleavage and subjected to MS2 mass spectrometry analysis. The fragmentation pattern is consistent with a single cleavage site between the galactose and gulose, designating RbmB as an α-1,4 lyase (red box indicates resulting tetrasaccharide). (F) Integration of peak intensity for the tetra- and octasaccharide over time (ON = overnight). The tetramer: octamer peak intensity ratio (red) remains relatively constant throughout the digestion. All data are represented as the mean ± SD ($n$ = 3).

mass of the terminal sugar by 2 Da, S4 Fig). The MS2 fragmentation pattern of RbmB-digested VPS was consistent only with cleavage of the α-1,4 linkage between the galactose and L-gulose monosaccharides, supporting our earlier observations (Fig 4E and S2 Table).

## RbmB is likely an endoglycosidase (lyase) that recognizes a footprint larger than a tetrasaccharide

To determine the origins of the mixture of tetra- and octasaccharides observed in the mass spectrometry analysis, we analyzed RbmB cleavage products as a function of time. Following the peak intensities of tetramer and octamer over a period of hours to overnight, the ratio of

tetramer to octamer does not vary significantly (Fig 4F). If RbmB recognized the tetrameric repeat and cleaved every galactose-gulose linkage, we would expect to observe a relative increase in the tetrameric species when the reaction is complete. One interpretation of the results is that the footprint recognized by RbmB is larger than a tetramer, leading to a pool of octasaccharides that cannot be further digested. Interestingly, we did not detect fragments larger than an octasaccharide and we did not observe ladders in our native gels (Fig 2A), which would be expected if there were appreciable levels of VPS fragments that could bind multiple Bap1 molecules at the same time.

We generated a simple computational model to further explore these results and to consider how the product outcome depends on the cleavage mechanism. In this model, each tetramer is represented as a unit, and a VPS polymer contains 1,000 such units. In each cycle, the polymer was cut into two pieces at a randomly chosen site representing cleavage by RbmB (S5 Fig). Four different cleavage strategies were simulated as follows: 1) RbmB binds to a polymer of length ≥2 units and cuts a unit from the end (exoglycosidase model), 2) RbmB binds to a polymer of length ≥2 units at a random 2-unit recognition site and cuts in the middle of the site, 3) RbmB binds to a polymer of length ≥2 units at a random 2-unit recognition site and cuts to the right of the site, and 4) RbmB binds to a polymer of length ≥3 units at a random 3-unit recognition site and cuts between the 2nd and 3rd units. For the first two strategies, the cleavage cycles were iterated until the pool consisted of only monomers (representing tetrasaccharides); and for the latter strategies, the simulation was terminated when the pool consisted of monomers and dimers (representing tetra- and octasaccharides).

Although the model is unable to shed light on the kinetics of cleavage and concentration-dependent phenomena, the possible outcomes in comparison with our results elucidate the possible enzymatic activity of RbmB. We find that the persistence of octamers observed in our mass spectrometry data is not consistent with strategies 1 and 2, each of which would eventually cleave VPS into only tetrameric species (S5A and S5B Fig). This prediction and our mass spectrometry data strongly suggest that RbmB is an endoglycosidase rather than an exoglycosidase and that it is unable to cut an octamer into two tetramers, as would occur in strategy 2. Our data, however, are consistent with strategies 3 and 4, both of which lead to a mixture of tetramers and octamers in the end (S5C and S5D Fig). When we look at the populations of n-mer species at successive times over a representative simulation, we never accumulate significant amounts of species larger than an octamer, except in the initial stages of the simulation (S5E and S5F Fig). This observation is consistent with the absence of detectable larger species in the mass spectrometry data. This may also explain why we do not see intermediately-sized native gel bands during RbmB cleavage arising from VPS fragments bound to multiple Bap1 molecules (Fig 2A).

## Solid-state and solution NMR of VPS confirms structure of cleavage product

To complement mass spectrometry-based detection and further characterize RbmB cleavage of VPS, we examined intact and RbmB-digested VPS by nuclear magnetic resonance (NMR) experiments. In particular, we sought to obtain additional atomic-level chemical detail towards understanding the enzymatic mechanism and activity of RbmB. We grew *V. cholerae* on nutrient agar medium containing $^{13}$C-labeled glucose as the only carbon source to enable uniform $^{13}$C-labeling and isolated VPS as in the previous experiments. The intact isolated VPS sample was examined by $^{13}$C cross-polarization magic-angle spinning (CPMAS) solid-state NMR [30], and the soluble cleavage products following RbmB cleavage of the $^{13}$C-labeled VPS were examined by solution NMR experiments.

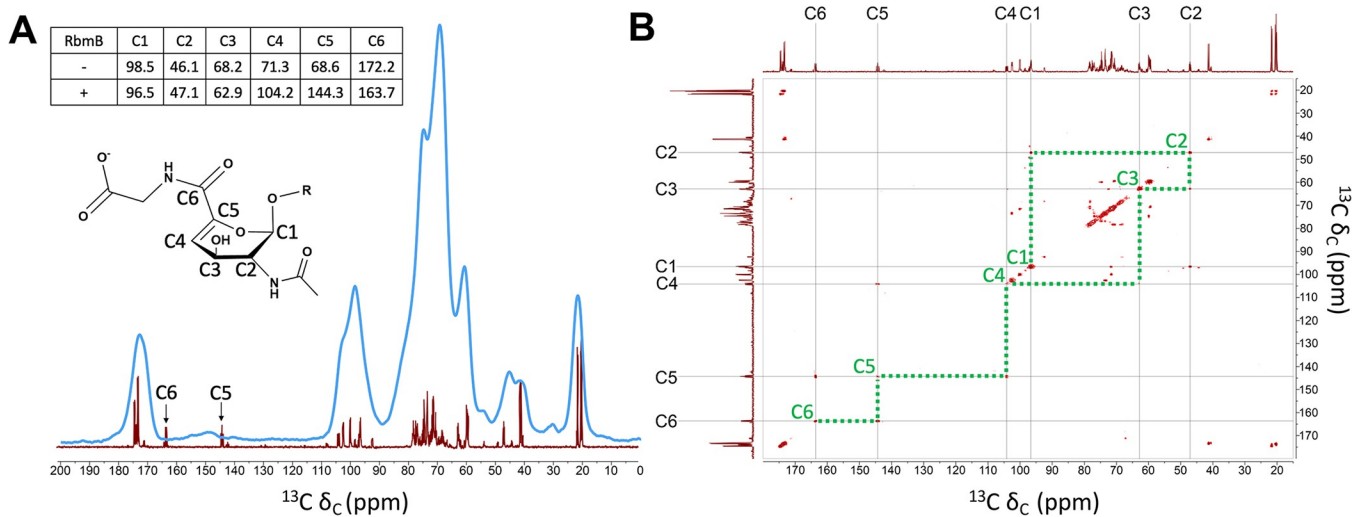

**Fig 5. NMR analysis of uncleaved and cleaved VPS.** (A) Superposition of 1D $^{13}$C NMR spectra collected on uncleaved VPS (using CPMAS solid-state NMR spinning at 10,000 Hz, blue) and RbmB-cleaved VPS (solution NMR, red) shows the presence of new peaks in the cleaved sample (carbons numbered as shown on proposed modified gulose structure). The chemical shifts of these two new peaks are consistent with formation of alkene carbons predicted to result from lyase activity on the gulose moiety. The inset table shows chemical shift values (in ppm) for uncleaved VPS as reported in [21] and cleaved VPS from the solution NMR data. (B) Solution 2D [$^{13}$C, $^{13}$C] COSY experiment run on RbmB-cleaved VPS shows assignments for carbons around the gulose ring. Green dotted lines illustrate the connectivity of carbon atoms around the gulose ring as labeled in (A).

A spectral overlay of the one-dimensional solid-state CPMAS spectrum of the full-length VPS and the solution-state spectrum of the cleaved VPS (Fig 5A) shows good overall correspondence of major chemical shifts between the two samples, as well as with published NMR spectra initially reported for the VPS structure determination [21]. The spectrum includes carbons associated with the carbohydrate C1 anomeric carbons (~100 ppm); the ring sugar carbons (60–85 ppm); carbonyls (~175 ppm) and methyls (~20 ppm) associated with acetylation; and glycine α-carbons (~40 ppm). Remarkably, and as we anticipated, we observed a major difference between the two spectra on top of the spectral broadening inherent to solid-state NMR—the appearance of two new peaks in the solution NMR spectrum of the cleaved VPS, consistent with new alkene carbons in the gulose moiety at ~144 ppm and the shifted C6 carbonyl carbon to ~164 ppm. We additionally performed a $^{13}$C-$^{13}$C correlation spectroscopy (COSY) experiment on the RbmB-cleaved gulose to attempt to specifically detect the coupling between the C4 and C5 alkene carbons and the C5 (alkene) and C6 (carbonyl) carbons as illustrated in the structure (Fig 5B). Indeed, cross-peaks between C4 and C5 and between C5 and C6 can be clearly identified. Thus, the comparative NMR data support RbmB's identification as a lyase resulting in the generation of a double bond between C4 and C5, in conjugation with the C6 carbonyl.

### *In vivo* activity of recombinant RbmB against *V. cholerae* biofilms

Following the extensive characterization of RbmB's biochemical activity with VPS substrates *in vitro*, we moved to biofilm culture to investigate how RbmB influences *V. cholerae* biofilm structure and dynamics. *V. cholerae* biofilm dispersal or disruption was evaluated using an imaging-based assay for biofilm growth and quantification [20,31], with *V. cholerae* grown in wells of glass-bottomed 96-well microtiter plates and quantification of well-associated biomass remaining after nutrient limitation and medium washes. We nutrient-limited cells to trigger dispersal in wild-type biofilms and to slow growth for the quantification of biofilm mass. To

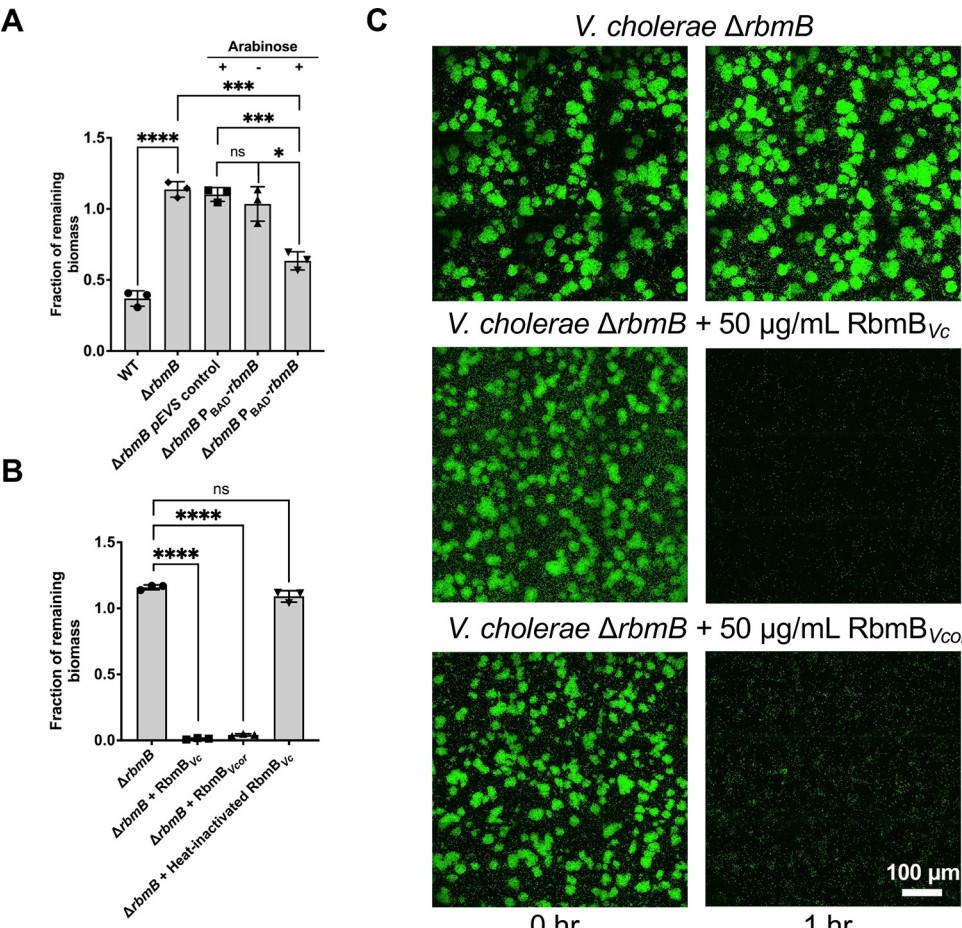

**Fig 6. RbmB disrupts *V. cholerae* biofilms.** (A) After biofilm growth in M9 medium with 0.5% glucose and 0.5% casamino acids as nutrient source, we induced nutrient limitation by removing the carbon sources in the medium. Biomasses before and after dispersal for 5 hours at 30˚C were quantified using a confocal microscope. The Δ*rbmB* mutant retains more surface biofilm mass following incubation, suggesting a weaker dispersal phenotype. Induction of *rbmB* from a plasmid under arabinose control shows partial recovery of the dispersal phenotype. Statistical analyses were performed using an unpaired, two-tailed *t*-test with Welch's correction. ns = not significant, *p < 0.05, ***p < 0.001, ****p < 0.0001. (B) Recombinantly purified RbmB (50 µg/mL) from *V. cholerae* (*Vc*) or *V. coralliilyticus* (*Vcor*) is sufficient to nearly completely disrupt *V. cholerae* biofilms within 1 hour, but heat-inactivated RbmB has no effect. All data are depicted as mean ± SD. Statistical analyses were performed using an unpaired, two-tailed *t*-test with Welch's correction. ns = not significant, ****p < 0.0001. (C) Cross-sectional images of *V. cholerae* biofilms at the bottom cell layer, before and after 1 hour of nutrient limitation. 50 µg/mL RbmB from *V. cholerae* (*Vc*) or *V. coralliilyticus* (*Vcor*) were present during the nutrient limitation step, in the middle and bottom rows, respectively. All data in this figure are depicted as mean ± SD (*n* = 3 biologically independent samples).

first investigate the influence of endogenous RbmB on biofilm dynamics, we generated a *V. cholerae* strain in which the gene for RbmB is deleted. We confirmed that deletion of *rbmB* leads to a significant defect in the dispersal assay, consistent with prior reports [20,31], and we were able to rescue the phenotype with a plasmid in which *rbmB* expression was induced with arabinose (Fig 6A).

Next, we assayed the ability of exogenously added RbmB to cause biofilm disruption in the Δ*rbmB* mutant biofilm, which allows us to avoid the confounding effect of natively produced RbmB. Incubation with 50 µg/mL of recombinant RbmB for 1 hour was sufficient to disrupt nearly all of the well-associated biomass as compared to the control without addition of RbmB (Fig 6B), revealing that the recombinant RbmB is active against living *V. cholerae* biofilms. As

a control, heat-inactivated RbmB is unable to produce any effect. To test whether nutrient limitation is necessary for biofilm disruption by RbmB, we repeated the assay using the Δ*rbmB* strain, but without nutrient limitation. Recombinant RbmB was equally able to disrupt biofilms in the presence of nutrients, while heat-inactivated RbmB was not (S6A and S6B Fig). To further examine the activity of recombinant RbmB, we followed biofilm disruption over time using two concentrations of purified RbmB and without nutrient limitation in the wild-type background (S6C Fig). We see a dose-dependent amount of disruption of wild-type *V. cholerae* biofilms over time, suggesting that RbmB is sufficient for disruption activity independent of nutrient limitation.

To address whether RbmB-induced biofilm disruption is directly related to VPS digestion, we used fluorescently labeled wheat germ agglutinin (WGA) to directly stain VPS (due to the GlcNAc component of VPS [16]) and imaged biofilms with or without added recombinant RbmB. Whereas mutant Δ*rbmB* biofilms showed no reduction in the VPS staining signal post 1 or 5 hours, addition of recombinant RbmB caused a loss of VPS staining signal in a time-dependent manner (S7 Fig). We saw a similar effect with wild-type *V. cholerae* biofilms (S7 Fig). The reduction in VPS staining signal is consistent with the disruption of the biofilm structure (Figs 6 and S7).

We also produced a recombinant RbmB in the *E. coli* expression system corresponding to that found in the coral-pathogen *Vibrio coralliilyticus*. RbmB in *V. coralliilyticus* and *V. cholerae* share approximately 47% similarity in protein sequence. *V. coralliilyticus* RbmB was similarly able to cleave VPS produced by *V. cholerae in vitro* (S8 Fig) and disrupt *V. cholerae* biofilms (Fig 6B and 6C) in the biofilm assays, indicating that both biofilm-forming species may produce a similar VPS or that the RbmB enzymes are promiscuous between the two species. These results suggest that RbmB is involved in biofilm dispersal when produced endogenously in *V. cholerae*, and that it can promote biofilm disruption when added exogenously.

## Discussion

VPS is an unusual polymer with a multiply-modified L-gulose sugar and a repeating pattern of α- and β-glycosidic bonds. Using a combination of biochemical, analytical, and structural tools, we demonstrate that *V. cholerae* produces an endoglycosidase, RbmB, that cleaves VPS between the gulose and galactose moieties to yield a mixture of tetrameric and octameric fragments. In addition, using a biofilm assay we showed that recombinantly-produced RbmB can facilitate *V. cholerae* biofilm disruption with or without nutrient limitation. Similar to other carbohydrate substrates containing uronic acid sugars (such as alginic acid or pectin), we hypothesize that RbmB mediates bond breaking in VPS through a β-elimination lyase mechanism, leading to a double bond between the C4 and C5 positions in the gulose moiety which was observed by solution NMR of RbmB-cleaved VPS [32]. Furthermore, the AlphaFold2-predicted structure of RbmB suggests a parallel β-helix fold, similar to other lyases including the PL6 family of alginate lyases, which also utilize calcium in their enzymatic mechanism [33]. Despite these similarities, VPS and alginate differ in their composition, with alginate containing a complex mixture of 1,4-linked β-D-mannuronic and α-L-guluronic acids. Future work is necessary to understand the differing enzymatic specificities of RbmB and alginate lyases, and to uncover structural details of RbmB's active site and how it recognizes and cleaves the unique VPS molecule.

Beyond *V. cholerae*, the VPS production system shares similarities with other polysaccharide secretion systems [34]. VPS production is encoded by a series of approximately 18 genes [12,35] with similarities to the Wzy/Wzx-dependent system of capsule synthesis and export

found in *E. coli* [36,37] and other exopolysaccharide-producing bacteria. In terms of biological activity, we demonstrate that recombinant RbmB from both *V. cholerae* and *V. coralliilyticus* species can disrupt *V. cholerae* biofilms grown in the laboratory. This raises questions about the specificity of RbmB against VPS polymers from different biofilm-forming Vibrio species (many contain RbmB orthologs) and whether they make a chemically similar VPS [34].

While the presence of a predicted signal peptide in RbmB suggests that it might localize to the periplasm or be further secreted [15], a mechanism for how RbmB leads to dispersal has been elusive. Cleaving VPS into smaller fragments would be an efficient way to disassemble the biofilm matrix, particularly in concert with other secreted matrix-digestion proteases that target other accessory matrix proteins, as has been previously demonstrated for *V. cholerae* [20]. Alternatively, RbmB could be involved in the release of VPS from the cell. Further insights into the mechanism and specificity of RbmB is necessary for understanding *V. cholerae* biofilm dispersal, as well as the potential for utilizing similar systems for the treatment of difficult biofilm-dependent infections [38,39] or as starting materials for designing digestible carbohydrate-based biomaterials [40]. A toolbox of glycosidic enzymes may also be useful for generating fragments of complicated natural oligosaccharides as an alternative to complex chemical synthesis.

## Materials and methods

### Strains used in this study

NEB5α *E. coli* (C2987I) and T7 Express *E. coli* (C2566I) were obtained from New England Biolabs, Ipswich, MA. The *V. cholerae* strain used in this study was a derivative of the WT *V. cholerae* O1 biovar El Tor strain C6706str2 (Table 1).

### Cloning of RbmB

The gene fragment coding for *V. cholerae* RbmB (VC0929) amino acids 32–377 were amplified using primers 5' GCG CGC CTC GAG TCA ATC TTT AAT AAA GTG CTG and 5' GCG CGC AGT ACT GAG GTT AAC TCT CAC AAT GTC ATA G from genomic DNA (wild-type *V. cholerae* O1 biovar El Tor strain C6706str2). The gene fragment coding for amino acids 32–370 of *V. coralliilyticus* RbmB (WP_021457174) was amplified using primers 5' GCG CGC CTC GAG CTA ATT TCC ATC GAC CTT TAG CGT TTT G and 5' GCG CGC AGT ACT TCT CAG GAC AAG GTG TAC CGA CTG from genomic DNA (OCN008). The PCR product was cloned into pET39b (Novagen) using the restriction sites for ScaI and XhoI (underlined).

**Table 1. Bacterial strains used in this study.**

| Strain Name in Paper | Genotype and Antibiotic Resistance | Description | Strain# & Reference |
|---|---|---|---|
| WT | Δ*VC1807::P_tac-mNeonGreen*, Spec^R | Transformation with Δ*VC1807::P_tac-mNeonGreen*, Spec^R into a WT strain | JY447 |
| Δ*rbmB* | Δ*rbmB*, Δ*VC1807::P_tac-mNeonGreen*, Spec^R | Clean deletion of *rbmB* in the WT background | JY552 [31] |
| Δ*rbmB*+P_BAD | *pEVS* (*P_BAD*, Kan^R), Δ*VC1807::P_tac-mNeonGreen*, Spec^R | *pEVS* control plasmid was mated into JY552 | XH159 |
| Δ*rbmB*+P_BAD-*rbmB* | *pEVS* (*P_BAD_rbmB*, Kan^R), Δ*VC1807::P_tac-mNeonGreen*, Spec^R | *pEVS* containing *rbmB* with an arabinose inducible promoter was mated into JY552 | XH160 |
| Rugose Δ*rbmA*Δ*bap1*Δ*rbmC*Δ*pomA* | *vpvC*^W240R Δ*rbmA*Δ*bap1*Δ*rbmC*Δ*pomA* | For VPS purification | JY286 [41] |

## Protein expression and purification

Expression plasmids containing either *V. cholerae* or *V. coralliilyticus rbmB* were transformed into T7 Express *E. coli* cells (NEB) and cultured overnight at 37˚C in LB media supplemented with 50 μg/mL kanamycin. Expression was performed in 500 mL cultures in high-cell density growth medium [42]. Cultures were inoculated with a 1:100 volume of overnight cells and grown at 37˚C until the $OD_{600}$ reached 0.3 to 0.5, at which point the cultures were induced with 250 μM of IPTG. An additional 18 hours of growth at 18˚C was allowed following induction before harvesting. Cells were harvested by centrifugation at 5,422 x g in a Sorvall LYNX 6000 centrifuge using a F9-6x1000 LEX rotor. Cell pellets were resuspended in 10 mL of lysis buffer (20 mM Tris pH 8.0, 35% (w/v) sucrose, 500 mM NaCl, 10% glycerol). For purification, cells were homogenized by passing three times through an Avestin EmulsiFlex-C5 high-pressure homogenizer and lysate was cleared by centrifugation at 29,000 x g in a Sorvall LYNX 6000 centrifuge using a F20-12 x 50 LEX rotor. DsbA-RbmB was purified from the cleared lysate by loading the supernatant onto a cobalt-charged 5 mL His-trap HF (GE Healthcare) equilibrated with high-salt TBS buffer (20 mM Tris pH 7.6, 500 mM NaCl). The column was washed with 10 column volumes of buffer containing high-salt TBS + 40 mM imidazole and the protein eluted in high-salt TBS buffer containing 250 mM imidazole. The protein was further purified using a Superose S6 100/300 (Cytiva) size exclusion chromatography column in high-salt TBS buffer (20 mM Tris pH 7.6, 500 mM NaCl). Fractions containing the purified protein were identified by SDS-PAGE and concentrated using a 50 kDa Amicon centrifugal concentrator.

The DsbA tag was cleaved using a 1:1000 w/w ratio of human α-thrombin (Haematologic Technologies Inc.) by incubation for 1.5 hrs at room temperature and arrested with 1 mM AEBSF (GoldBio) and 10 mM EDTA. Cleaved RbmB was separated from DsbA by purification on a Superdex 75 100/300 (Cytiva) size exclusion column coupled to a 1 mL His-trap HF column (GE Healthcare) equilibrated with 20 mM HEPES and 500 mM NaCl. Fractions containing cleaved RbmB were determined by SDS-PAGE and concentrated using a 10 kDa Amicon centrifugal concentrator. Recombinant Bap1-GFP$_{UV}$ was expressed and purified as previously described [43]. Free DsbA was purified in a similar manner by expressing empty pET39b vector (without the RbmB insert) and purifying the protein as described above using the polyhistidine tag.

## VPS purification

VPS purification was performed according to a published protocol with several modifications [18]. First, a rugose Δ*rbmA*Δ*bap1*Δ*rbmC*Δ*pomA* strain was grown in LB at 30˚C overnight. 50 μL of this inoculum was added into 3 mL of LB liquid medium containing glass beads, and the cultures were grown with shaking at 30˚C for 3–3.5 h. 50 μL of this inoculum was applied to an agar plate containing M9 medium with 0.5% glucose and 0.5% casamino acids and were incubated at 30˚C for 2 days to form a continuous bacterial lawn. For each purification batch, 20 plates were used. The biofilms were scraped off the agar plates carefully and resuspended in 1× PBS. Biofilm cells were removed by centrifugation (8000 × g, 4˚C, 45 min) and the supernatant was dialyzed for 3 days against distilled water using a dialysis cassette (10 kDa MWCO) with repeated water changes. The dialyzed sample was lyophilized to prepare crude VPS extract. The crude extract was dissolved in 10 mM Tris buffer at 1.5 mg/mL, treated with DNase and RNase (37˚C, 24 h), and then Proteinase K (37˚C, 48 h), followed by ultracentrifugation at 100,000 × g for 1 h to remove lipopolysaccharide. This solution was dialyzed against water for 3 days and lyophilized to produce VPS. For each purification batch, typically 10–15 mg of VPS was obtained as a white powder after the final lyophilization step. The VPS solutions were heated at 95˚C for 10 min to denature the Proteinase K before use.

## Native-PAGE VPS digestion analysis

Samples for RbmB digestions (using uncleaved DsbA-RbmB) were set up in 1x TBS buffer (20 mM Tris-HCl, pH 7.8, 150 mM NaCl) at amounts of 0, 0.5, 1, 2.5, 5, 10, and 15 μg and a VPS amount of 5 μg per well/lane. Following incubation overnight, samples were boiled for 5 min to denature RbmB and quench the reaction. Digested VPS samples were preincubated with 5 μg of Bap1-GFP$_{UV}$ for 5 min before running on an 10% acrylamide native-PAGE gel with a 5% stacking gel. The native running buffer contained 25 mM Tris-HCl, pH 8.3 and 192 mM glycine. The native loading buffer was made with 62.5 mM Tris-HCl, pH 6.8, 25% glycerol, and 1% bromophenol blue dye. Gel electrophoresis was performed at 85 V for 4 h in an ice bath. Images were acquired on gels (encapsulated in glass plates) by imaging the Bap1-GFP$_{UV}$ fusion using an excitation wavelength of 492 nm and an emission wavelength of 513 nm on a Typhoon FLA 9000 imaging system (GE Healthcare). Quantification of Bap1-GFP$_{UV}$ bands was performed using ImageQuant TL analysis software (Cytiva).

## RbmB activity assays

The *in vitro* VPS digestion activity of DsbA-RbmB was quantified using an ethanolamine-borate fluorescence assay [44,45]. Reactions were incubated at 30°C using 20 mM HEPES and 150 mM NaCl buffer and quenched by heating to 95°C for 5 mins followed by centrifugation at 21,000 x g for 5 mins. Following cleavage, reaction mixtures were adjusted to 50 μL using 20 mM HEPES and 150 mM NaCl buffer. Supernatants were added to a glass tube containing 300 μL ethanolamine-borate reagent (4.67 g boric acid (Thermo-Fisher), 4.67 mL ethanolamine (Sigma), 100 mL ddH$_2$O; passed through a 0.22 μm filter). Samples were incubated in a heat block at 120°C for 15 minutes and their fluorescence measured using a FluoroMax 4 (Horiba) fluorescence spectrometer using an excitation wavelength of 357 nm and an emission wavelength of 443 nm. A standard curve of 0 to 600 μM was prepared using galactose to quantitatively determine the number of reducing ends per sample. For all activity assays, 25 μg of VPS and 20 μg of RbmB were used and the reactions incubated at 30°C for 30 mins. The optimal salt concentration was determined by incubating in 20 mM HEPES with NaCl concentrations of 150, 250, 350 and 500 mM. All other assays were performed in 20 mM HEPES and 150 mM NaCl buffer, under standard assay conditions. The optimal temperature for activity was determined by incubating over a temperature range of 18 to 50°C. The dependence of activity on pH was determined by incubating the enzyme in Teorell and Stenhagen buffer (as described in [46]), with a pH range of 3–11 (330 mM citric acid monohydrate, 330 mM phosphoric acid, 167 mM boric acid). The effect of metal ions on activity was determined by incubating in buffer solutions containing CaCl$_2$, MgCl$_2$, EGTA or EDTA at a final concentration of 10 mM. Optimization data for pH and metal ions were acquired by incubating at 30°C for 30 mins, followed by inactivation of the enzyme at 95°C for 5 min.

RbmB specificity was determined by testing against VPS, sodium alginate (Sigma), hydroxyethyl cellulose (Sigma), arabinan (Carbosynth), and β-D-glucan (Carbosynth) using the method described above. 25 μg of each polymer was incubated with 20 μg of RbmB at 30°C for 60 mins. Alginate lyase (Sigma) was used to digest alginate using the same concentrations, but incubated at 37°C for 60 mins. All samples were prepared in 20 mM HEPES and 150 mM NaCl buffer.

## LC/MS analysis of cleaved VPS samples

RbmB-cleaved VPS samples (100 μg VPS with 80 μg RbmB overnight) were purified using a graphitized carbon solid-phase extraction (SPE) column (Sigma-Aldrich) to remove protein, buffer, and undigested polysaccharides. The SPE column was first activated with 3 x 500 μL

acetonitrile containing 0.1% trifluoroacetic acid (TFA), followed by equilibration with 3x 500 μL of 0.1% TFA solution in water. Samples were dissolved in 50–100 μL of 0.1% TFA, loaded onto the column, washed with 3 x 500 μL water with 0.1% TFA, and then purified oligosaccharides were eluted with 3 x 500 μL 50% acetonitrile with 0.1% TFA, followed by centrifugal evaporation until the samples were completely dry. For reducing end identification, a sample was incubated at 50˚C in 50 mM KOH and 1 M $NaBH_4$ for 3 hours, and then carbon extraction proceeded as described earlier.

Prior to acquisition by mass spectrometry (Orbitrap Fusion Lumos Tribrid mass spectrometer, Thermo Fisher Scientific), samples were dissolved in Milli-Q water. For infusion analysis, the sample was directly infused into an operating spectrometer in both positive and negative polarities. Data were acquired for both the precursor (MS1) and fragments (MS2) to confirm the exact mass and polysaccharide sequence, respectively. For liquid chromatographic separation, the digested sample was run on a porous graphitized carbon (PGC) HPLC column packed in-house in fused silica (100 mm x 75 μm, 3-micron Hypercarb particles, Thermo Fisher Scientific) held at 75˚C, which separates digested polysaccharides using reverse-phase chromatography. An Ultimate 3000 capillary flow pump delivered the mobile phases, composed of 10 mM ammonium bicarbonate in water (Buffer A) and 10 mM ammonium bicarbonate in 80% acetonitrile, 20% water (Buffer B), using a linear gradient from 4% - 25% B over 35 min, followed by a 5 min wash at 99% B, then 5 min equilibration at 4% B.

Acquired data were analyzed in a semi-automated manner, using GlycoWorkbench [47] and previous characterization studies for monosaccharide stereochemistry assumptions. Spectrum visualization was performed in Freestyle 1.8 SP2 (Thermo Fisher Scientific). The molecular structure, chemical formula, molecular mass, and molecular weight of VPS and VPS variants were displayed or determined using ChemDraw (Revvity Signals).

## *V. cholerae* biofilm dispersal or disruption assays

*V. cholerae* strains constitutively expressing mNeonGreen were grown overnight in LB at 37˚C. 50 μL from each culture was used to inoculate 1.5 mL of M9 medium supplemented with 0.5% glucose and 0.5% casamino acids and grown at 30˚C with shaking until the $OD_{600}$ was between 0.1 and 0.3. The cultures were then diluted to an $OD_{600}$ of 0.03. 1 μL of regrown culture was added to 100 μL M9 0.5% glucose and 0.5% casamino acids and incubated at 25˚C for 20 hours (Δ*rbmB*) or 12 hours (WT); we had to use different timing because the WT biofilm starts to disperse spontaneously at an earlier time [20]. The well was washed twice with M9 0.5% glucose/0.5% casamino acids, replaced by the same medium, and imaged with a spinning disk confocal microscope (Nikon Ti2-E connected to Yokogawa W1) using a 60× water objective (numerical aperture = 1.20) and a 488 nm laser excitation. For each sample, several locations with 3×3 tiles were imaged and captured with an sCMOS camera (Photometrics Prime BSI). The x-y pixel size was 0.22 μm and the z-step size was 3 μm. The medium was then taken out, replaced by M9 media without any carbon source (nutrient limitation) or M9 0.5% glucose/0.5% casamino acids (no nutrient limitation), and incubated for a period of time (5 hours for data in Fig 6A, 1 hour for data in Figs 6B and S6A, or 1/3/5 hours for the time course in S6C Fig) at 30˚C. The wells were then washed twice with M9 medium (nutrient limitation) or M9 0.5% glucose/0.5% casamino acids (no nutrient limitation) and re-imaged at the same locations. For the complementation assay, all growth media additionally contained 100 μg/mL kanamycin.

For heat-inactivated RbmB, 0.5 mg/mL RbmB was incubated at 95˚C for 10 minutes before diluting with corresponding media (M9 media without any carbon source or M9 0.5% glucose/0.5% casamino acids). All biofilm experiments were conducted with the uncleaved DsbA-RbmB fusion protein.

Image analysis was performed with built-in functions of the Nikon Elements software by thresholding each image layer-by-layer and measuring the total binarized area above the threshold in each layer. The binary area for each sample z-slice was then summed to give the total biovolume, and the ratio of the total biovolume after versus before the treatment step was calculated.

## VPS staining and imaging

*V. cholerae* strains constitutively expressing mNeonGreen were grown overnight in LB at 37˚C. 50 µL from each culture was used to inoculate 1.5 mL of M9 medium supplemented with 0.5% glucose and 0.5% casamino acids and grown at 30˚C with shaking until the $OD_{600}$ was between 0.1 and 0.3. The cultures were then diluted to an $OD_{600}$ of 0.03. 1 µL of regrown culture was added to 100 µL M9 0.5% glucose and 0.5% casamino acids containing 0.5 mg/mL BSA (Sigma-Aldrich) and 4 µg/mL WGA-AlexaFluor647 (Sigma-Aldrich) and incubated at 25˚C for 20 hours (Δ*rbmB*) or 12 hours (WT). The well was washed twice with M9, replaced by the same medium, and imaged with a spinning disk confocal microscope (Nikon Ti2-E connected to Yokogawa W1) using a 60× water objective (numerical aperture = 1.20) and 488 nm and 640 nm laser excitation. For each sample, several locations with 3×3 tiles were imaged and captured with an sCMOS camera (Photometrics Prime BSI). The x-y pixel size was 0.22 µm and the z-step size was 3 µm. The medium was then taken out, replaced by M9 media containing 0.5 mg/mL and 4 µg/mL WGA-AlexaFluor647, and incubated for 1 or 5 hours. The wells were then washed twice with M9 medium and re-imaged at the same locations.

## Detection of VPS cleavage by EM

Digested VPS was produced by incubating 5 µg of VPS with 4 µg of RbmB at 30˚C, overnight. 2 µg of Bap1, 1 µg of VPS, and 1 µg of digested VPS were prepared in a total volume of 20 µL, and incubated for 5 mins. Prior to grid preparation, reactions were diluted in 1x TBS.

Copper grids with formvar support film (Ted Pella, Inc.) were glow-discharged for 5 mins and 4 µL of the diluted sample was spotted onto grids and incubated for 5 mins. Grids were stained 3 times with 2% uranyl acetate for 5–10 seconds per drop, in series. Excess stain was removed by filter paper and grids incubated at room temperature for 10 mins. Grids were screened using a Thermo Scientific Talos L120C transmission electron microscope.

## Simulation of VPS digestion

Cleavage of VPS polymer by RbmB was simulated using a custom script written in Python 3. Beginning with a pool consisting of one 1000-mer (where each unit represents a tetrameric unit of VPS), the enzyme iteratively binds to and cuts a randomly chosen polymer in the pool into two fragments, until the polymer pool is exhausted. Every polymer is equiprobable to be chosen from the pool in each iteration, regardless of length, and four strategies for binding and cutting were tested:

1. Bind to a polymer of length $\geq 2$ and cut a monomer from the end.

2. Bind to a polymer of length $\geq 2$ and cut a randomly chosen dimer within the polymer in the middle. Here, every dimer within the polymer is equiprobable to be chosen.

3. Bind to a polymer of length $\geq 2$ and cut a randomly chosen dimer within the polymer to the right (after the 2nd monomer). Every dimer within the polymer is equiprobable to be chosen (binding to the rightmost end of the polymer or to a dimer leads to no cutting for that cycle).

4. Bind to a polymer of length $\geq 3$ and cut a randomly chosen trimer within the polymer between the 2nd and 3rd monomers. Every trimer within the polymer is equiprobable to be chosen (each cycle yields a cut).

In strategies 1, 2, and 4, every binding event yields a cut. In strategy 3, if the bound polymer is in fact a dimer, or the enzyme chooses to cut the rightmost dimer of the bound polymer, then the binding event does not yield a successful cut.

For strategies 1 and 2, the simulation terminates when the pool consists only of monomers; for strategies 3 and 4, the simulation terminates when the pool consists only of monomers and dimers.

## Solid-state CPMAS NMR analysis of intact $^{13}$C-labeled VPS and solution NMR analysis of $^{13}$C-labeled VPS digested by RbmB

Uniformly $^{13}$C-labeled VPS was produced and isolated as described above with the following changes. Biofilms were grown on agar plates containing M9 medium with 0.5% $^{13}$C-labeled glucose (Cambridge Isotope Laboratories, Inc.) as the only carbon source. For solution NMR, 1 mg of $^{13}$C-labeled VPS was digested using 800 μg RbmB at 30°C, overnight. Digested samples were SPE extracted and lyophilized using the protocol mentioned previously.

Solid-state $^{13}$C CPMAS NMR experiments were carried out in an 89-mm-bore 11.7T magnet using a home-built four-frequency $^{1}$H/$^{19}$F/$^{13}$C/$^{15}$N all-transmission-line probe with a Varian VNMRS console. Samples were spun at 10,000 Hz in a thick-walled 5-mm-outer-diameter zirconia rotor. The field strength for cross-polarization (CP) was 50 kHz for $^{13}$C with ramped CP for $^{1}$H centered at 57 kHz (with 10% ramp) for 1.5 ms. $^{1}$H decoupling was performed with two-phase pulse-modulation (TPPM) [48] at 83 kHz. The recycle time was 2 s. Spectrometer $^{13}$C chemical shift referencing was performed by setting the high-frequency adamantane peak to 38.5 ppm [49]. The $^{13}$C CPMAS spectrum was the result of 4,096 scans.

Solution NMR experiments were performed at 25–50°C on an 800 MHz Agilent DDR2 NMR spectrometer with a 5 mm HCN cold probe using standard OpenVnmrJ experiments/ pulse sequences (https://zenodo.org/doi/10.5281/zenodo.5550488), WET-1D, gHSQCAD, CARBON, and C13 gCOSY. 128 scans were used for all spectra except for gHSQCAD, which used 8 scans and 512 increments. The C13 gCOSY 2D used 256 increments. WET solvent suppression was used to decrease the residual water signal in the 1D proton spectrum.

## Statistical analysis

All statistical analyses were performed using GraphPad Prism software (Dotmatrics). Three independent tests were performed for each experiment and error bars represent standard deviations. Statistical significance in the biofilm disruption and enzymatic assays were determined using an unpaired two-tailed *t*-test with Welch's correction.

## Supporting information

**S1 Table. Table listing abundant peaks in LC/MS data (Fig 4D) with actual and predicted adduct masses.**
(TIF)

**S2 Table. HCD-MS/MS fragmentation mass list of [M+H+Na] precursor, m/z 785.32.** Fragmentation annotations follow the Domon-Costello nomenclature [50].
(TIF)

**S1 Fig. Quantification of Bap1-GFP_UV band from native gel-shift analysis (lower arrow band from Fig 2A).** An increase in the amount of free Bap1 is observed, consistent with digestion of the VPS polymer into smaller fragments. Data represent a single replicate.
(TIF)

**S2 Fig. Fluorescence-based quantification of reducing ends using galactose as a standard in the presence (red) and absence (blue) of 10 mM CaCl$_2$.** All data are represented as the mean ± SD ($n = 3$).
(TIF)

**S3 Fig. Specificity of *in vitro* cleavage of VPS by RbmB from *V. cholerae*.** Fluorescence-based quantification of reducing ends before (U) and after (D) digestion is shown for several polymers. HEC represents hydroxyethyl cellulose. All data are represented as the mean ± SD ($n = 3$). Statistical significance was determined using an unpaired, two-tailed *t*-test with Welch's correction. ns = not significant, $^*p<0.05$, $^{**}p<0.01$. Solid lines indicate which samples were compared in the statistical tests.
(TIF)

**S4 Fig. Negative-ion mode mass spectrometry analysis of cleaved VPS before and after borate reduction.** RbmB-digested VPS was incubated with 50 mM KOH, 1M NaBH$_4$ for 3 hours at 60˚C. Peaks indicate the expected shift of +2 Da.
(TIF)

**S5 Fig. Simulation of RbmB cutting events based on several digestion strategies.** (A) Exoglycosidase strategy with RbmB cutting a single tetrasaccharide from the end of the VPS polymer. $k = 1$ and $k = 2$ represent the tetra- and octasaccharide species, respectively. This model predicts a linear accumulation of tetrasaccharide species over binding events. (B) Octasaccharide recognition strategy with cutting in the middle predicts primarily tetrameric species. (C) Octasaccharide recognition strategy with cutting at the end predicts a mixture of tetra- and octasaccharide species. (D) Dodecamer recognition strategy with cutting in the middle also predicts a mixture of tetra- and octasaccharide species. (E) Histogram showing accumulation of $k = 1$ and $k = 2$ species over time from a representative simulation in panel (C). (F) Histogram associated with panel (D).
(TIF)

**S6 Fig. Nutrient limitation is not necessary for biofilm disruption caused by exogenously added RbmB.** (A) Biomasses of biofilms grown without nutrient limitation before and after introduction of RbmB (at 50 μg/mL) for 1 hour at 30˚C were quantified using a confocal microscope. Heat-inactivated RbmB (50 μg/mL) had no significant effect on biofilm disruption. (B) Relative enzymatic activity of 20 μg of recombinant RbmB against 25 μg of purified VPS for 30 min demonstrates a loss of activity following heat-inactivation. Cleavage and removal of the DsbA tag does not alter the RbmB activity and purified DsbA is unable to cleave VPS on its own. (C) Wild-type *V. cholerae* biofilms grown without nutrient limitation are disrupted in a dose-dependent manner by recombinant RbmB. All data are depicted as mean ± SD. Statistical analyses were performed using an unpaired, two-tailed *t*-test with Welch's correction. ns = not significant, $^{***}p < 0.001$, $^{****}p < 0.0001$. Solid lines indicate which samples are compared for each statistical test.
(TIF)

**S7 Fig. Biofilm imaging using wheat germ agglutinin (WGA) shows a loss of VPS following RbmB incubation.** Biofilms were imaged with fluorescently labeled WGA to stain VPS at 0 hours, 1 hour, and 5 hours. The WGA signal remained strong in the Δ*rbmB* mutant biofilms

(top row), while addition of 50 μg/mL RbmB led to a decrease in the WGA signal (middle row). The lower row shows the same experiment with the wild-type strain. The left two panels show cells (mNeonGreen fluorescence) and the right two panels show WGA staining before and 5 hours after addition of RbmB.
(TIF)

**S8 Fig. *In vitro* cleavage of VPS by RbmB from *V. cholerae* (*Vc*) and *V. coralliilyticus* (*Vcor*).** Fluorescence-imaged native-PAGE gel using Bap1-GFP$_{UV}$ fusion to image VPS polymer. Digestion of VPS with RbmB from either species leads to a reduction in Bap1/VPS aggregates.
(TIF)

## Acknowledgments

Negative stain electron microscopy data were collected at the Yale CryoEM resource (Science Hill site) with help from Dr. Kaifeng Zhou. Mass spectrometry analysis was performed at Beth Israel Deaconess Medical Center—Glycomics Core, RRID:SCR_024818 (https://www.bidmc.org/research/core-facilities/glycomics-core). We would like to thank Dr. Lei Li and Prof. Fitnat Yildiz for helpful discussions, and Zane Lombardo for technical assistance.

## Author Contributions

**Conceptualization:** Jing Yan, Rich Olson.

**Data curation:** Ranjuna Weerasekera, Xin Huang, Yun Huynh, Christopher Ashwood, Lauren E. Pepi, Eric Paulson, Lynette Cegelski, Jing Yan, Rich Olson.

**Formal analysis:** Ranjuna Weerasekera, Alexis Moreau, Xin Huang, Kee-Myoung Nam, Xinyu Liu, Christopher Ashwood, Lauren E. Pepi, Eric Paulson, Lynette Cegelski, Jing Yan, Rich Olson.

**Funding acquisition:** Rich Olson.

**Investigation:** Ranjuna Weerasekera, Alexis Moreau, Xin Huang, Kee-Myoung Nam, Alexander J. Hinbest, Yun Huynh, Xinyu Liu, Christopher Ashwood, Lauren E. Pepi, Eric Paulson, Lynette Cegelski, Jing Yan, Rich Olson.

**Methodology:** Ranjuna Weerasekera, Alexis Moreau, Alexander J. Hinbest, Yun Huynh, Christopher Ashwood, Lauren E. Pepi, Eric Paulson, Lynette Cegelski, Jing Yan, Rich Olson.

**Project administration:** Jing Yan, Rich Olson.

**Resources:** Alexis Moreau, Jing Yan, Rich Olson.

**Software:** Kee-Myoung Nam, Jing Yan.

**Supervision:** Jing Yan, Rich Olson.

**Validation:** Ranjuna Weerasekera, Christopher Ashwood, Lauren E. Pepi, Lynette Cegelski, Jing Yan, Rich Olson.

**Visualization:** Ranjuna Weerasekera, Kee-Myoung Nam, Christopher Ashwood, Eric Paulson, Lynette Cegelski, Jing Yan, Rich Olson.

**Writing – original draft:** Rich Olson.

**Writing – review & editing:** Ranjuna Weerasekera, Alexis Moreau, Xin Huang, Kee-Myoung Nam, Alexander J. Hinbest, Christopher Ashwood, Lauren E. Pepi, Eric Paulson, Lynette Cegelski, Jing Yan, Rich Olson.

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
