## [Decision Letter · Decision Letter 0]

12 Aug 2024

Dear Dr. Olson,

Thank you very much for submitting your manuscript "Vibrio cholerae RbmB is an α-1,4-polysaccharide lyase with biofilm-dispersal activity against Vibrio Polysaccharide (VPS)" for review by PLOS Pathogens. Your manuscript was fully evaluated by members of the editorial board and by three independent peer reviewers. In light of the reviews (below), we would like to invite the resubmission of a substantially revised version that takes into account the reviewers' comments.

The reviewers appreciated the attention to an important problem, but raised some serious concerns that must be addressed before we will consider a revised version of your study. Your revisions should address all the specific issues from each reviewer, paying particular attention to the following:

1) The role of calcium in RbmB activity (Reviewer 2).

2) The pH optimum of RbmB activity (Reviewer 2).

3) Controls to include other bacterial EPS (Reviewer 3).

4) Clarity of the experiments performed that included the exogenous supplementation of RbmB (Reviewer 3).

5) Clarity of experimental details (all three reviewers).

We cannot make any decision about publication until we have seen the revised manuscript and your response to the reviewers' comments. Your revised manuscript is also likely to be sent to reviewers for further evaluation.

Sincerely,

Jon Paczkowski

Academic Editor

PLOS Pathogens

D. Scott Samuels

Section Editor

PLOS Pathogens

Michael Malim

Editor-in-Chief

PLOS Pathogens

orcid.org/0000-0002-7699-2064

Reviewer's Responses to Questions

**Part I - Summary**

Reviewer #1: This manuscript presents data that elucidate the biochemical function of the protein RbmB, which has been implicated in the biofilm dispersal process in Vibrio cholerae. Previous work has shown that RbmB is a glycosidase, degrading the polysaccharide component of the V. cholerae biofilm matrix, thereby enabling dispersal. However, the precise biochemical mechanism by which RbmB processes the polysaccharide component of the biofilm matrix was previously unclear. The authors find that RbmB cleaves the polysaccharide at the �-1,4 linkage between D-galactose and a modified L-guluronic acid into a mixture of octamers and tetramers via a glycoside lyase mechanism. The chemical reaction is highly specific, as the authors find that RbmB is unable to cleave a variety of other polysaccharides. As a proof of in vivo relevance, the authors find that, in a strain that does not express rbmB, addition of purified RbmB to biofilms results in complete dispersal Overall, the work is thorough in its characterization of the biochemical function of RbmB and would be a useful reference for the field. However, I believe the points below are important to address for the clarity and readability of the manuscript.

Reviewer #2: The manuscript by Weerasekera et al., “Vibrio cholerae RbmB is an -1,4-polysaccharide lyase with biofilm-dispersal activity against Vibrio Polysaccharide (VPS),” is based on the characterization of RbmB hydrolase function and its role in V. cholerae biofilm dispersion. Using a combination of biochemical and analytical tools, the authors demonstrated that RbmB acts as a polysaccharide lyase by cleaving VPS at a unique site into a mixture of tetramers and octamers. Furthermore, as showing purified RbmB disperses V. cholerae biofilms, the findings suggest that RbmB may play a role in VPS degradation. Overall, the study was well constructed and performed. The paper would be strengthened if several issues would be addressed. Comments and concerns are given below.

Reviewer #3: This manuscript experimentally investigates the enzyme RbmB and its role in biofilm dispersal. They expressed and purified recombinant RbmB and demonstrated that it cleaves VPS, an important component of Vibrio cholerae biofilms. Through various biochemical and analytical techniques including mass spectrometry and NMR, they outline many of the molecular details of RbmB and how it cuts VPS. They show that RbmB can disperse living V. cholerae biofilms when added exogenously.

I enjoyed reading this paper, and believe it provides interesting and important information about a key V. cholerae enzyme. I believe it will be an important part of the V. cholerae literature. However, before I can recommend publication, there are two issues I believe must be dealt with.

**Part II – Major Issues: Key Experiments Required for Acceptance**

Reviewer #1: (No Response)

Reviewer #2: - Description of experimental conditions in the results section would benefit from more detailed descriptions, e.g. it was not readily apparent what was loaded onto the SDS gel (l. 177-182).

- Figure 3A and Figure S1: There seems to be a discrepancy in the scale of the graph. The author mentioned that a standard curve was made using 0 to 600 mM of galactose (L 522-523), but the scale of Fig 3A and Fig S1 is μM. Could you kindly provide some clarification on this?

- Figure 4D and Figure S3: #4 and #6 chemical formula, which are presented in Fig S3, are not shown in Fig 4D. Is it because of the small peak?

- Figure 6A and B: Please clarify the treatment (Y axis). In Figure 6A, does the treatment mean arabinose induction or nutrient limitation for inducing biofilm dispersion? The treatment in Figure 6B may imply the exogenous addition of RbmB or nutrient limitation. The author needs to clearly explain this in the Figure legend.

- Figure 6C: The authors need to check the magnification of each confocal image. It seems that the magnification is different for each image.

- L219-223, Figure 3E: Fig 3E indicates that RbmB cleavage activity is increased in the presence of calcium. However, at L 220, the author mentioned that the presence of calcium is not required for the activity as treatment with EDTA did not abolish the cleavage. However, EDTA chelates divalent ions other than calcium. The authors should consider using the calcium specific chelator EGTA. Also, regarding the author’s explanation, is there any evidence that VPS interacts with calcium?

- Figure 3E: The data showed that the baseline has activity. Why does it have cleavage activity?

- pH optimum. The authors used HEPES buffer with a pH range of 3 to 11 to determine the pH optimum of RbmB activity. However, the HEPES buffer only has a pH range of 6.8-8.2. I recommend either deleting the corresponding Figure 3B or repeating the experiments using buffers with overlapping pH ranges. Sigma Aldrich provides a handy list of useful pH ranges of various buffers, see Buffer Reference Center (sigmaaldrich.com)

Reviewer #3: 1. The authors describe RbmB as being specific. In particular, they write "This suggests that the specificity of the VPS/RbmB interaction is unlikely to be limited to Vc and may be shared with other Vibrio species that exhibit VPS-dependent biofilm formation (including Vibrio mimicus and Vibrio anguillarum)." However, I think this statement is stronger than has been proven. RbmB is tested against three other molecules, but is not tested against other bacterial EPS. If the action of RbmB is to specifically cleave VPS, it seems important to show that it does not cleave the EPS made by other bacteria. Conversely, it could be interesting if it acts specifically to EPS of many types. Note, I believe it may be possible to address this by better explaining the other molecules tested, and better qualifying the uncertainty of how specific RbmB truly is.

2. The authors show that RbmB can disperse Vibrio cholerae biofilms when added exogenously in sufficient amounts. However, it is not shown that V. cholerae do this in vivo. The authors alude to this fact in the last paragraph of the discussion. However, much of the rest of the manuscript reads as if this is a demonstrated fact, e.g., the sentence "Finally, we confirmed that the cleavage strategy was used by Vc biofilms to disperse upon nutrient limitation." This seems too strong as the RbmB is added exogenously, rather than secreted as a "strategy" used by the biofilm to disperse. If the authors can show that WT biofilms disperse much faster than DeltaRbmB biofilms, then I think they could make a strong claim. I will again note that a clearer description of what was done and what is and is not known could be sufficient here.

**Part III – Minor Issues: Editorial and Data Presentation Modifications**

Reviewer #1: • The results section entitled, “Recombinant-purified RbmB disrupts VPS/Bap1 aggregates,” is confusing as written (particularly lines 172-178 describing the gel shift experiment with Bap1). Ultimately, the results in Figure 3 more convincingly demonstrate the enzymatic activity of RbmB against the VPS, compared to the two panels in Figure 2. I recommend that the authors consider either re-writing the description of the experimental approaches for Figure 2, namely how the Bap1 gel shifting experiment works, or simply remove these results from the manuscript as they are somewhat redundant with Figure 3 and are less compelling. Furthermore, the EM results in Figure 2 would be more convincing if a quantification of the monomers vs aggregates was performed.

• What is DsbA and how does it help with the purification of RbmB? Without this information the logic in the first paragraph is difficult to follow.

• Figure 5 is a difficult figure to interpret/read for non-specialists. It would be ideal to make it more legible (increase font size and the y-label in panel B in upside down). Can you point non-specialists to features of the plots that are relevant (as in Figure 4).

• In Figure 6, the fluorescence images are blown-out/look of low quality. The authors should fix this. How was normalization performed in Figure 6B?

• In some areas of the manuscript language is rather informal for a final publication. A non-comprehensive list of specific examples include:

o Why not just type out V. cholerae and V. coralliilyticus, as is done for E. coli?

o Why do the authors repeatedly indicate milligram quantities of RbmB were purified (in the main text). This is fairly routine for recombinant protein purification and does not help the reader to interpret anything.

o The term “mostly completing” on line 211 is ambiguous.

• Miscellaneous typos/word choice suggestions. This is a non-comprehensive list and I strongly suggest the authors carefully check for phrasing.

o In general, the authors could be more confident in describing established results about RbmB. For instance, on line 78, “suspected” should be changed to “known” as this is a well-established result.

o Line 77-78, “the V. cholerae gene cluster” should be “the V. cholerae biofilm gene cluster”.

o Line 112, “RbmB” should be “rbmB”. It is not possible to delete a protein.

o Line 132 “uses” should be “use” and line 134, “results” should be “result”. In general, check that the correct verb tenses are used throughout the manuscript.

o Line 212, remove “on”

o Line 240, remove “obviously”

Reviewer #2: - Figure 2A: Quantification of the Bap1/VPS complex abundance in each well would be helpful in understanding the native gel experiment.

- L206 and Figure S1: Please clarify which lines are +/- CaCl2 in both the figure and figure legend.

- L272: Explain what are Gal-Gul.

- Please change Figure S3 and S5 to Table S1 and S2.

Reviewer #3: (No Response)

PLOS authors have the option to publish the peer review history of their article (what does this mean?). If published, this will include your full peer review and any attached files.

Reviewer #1: No

Reviewer #2: **Yes: **Karin Sauer

Reviewer #3: No
---

## [Decision Letter · Decision Letter 1]

15 Oct 2024

Dear Dr. Olsen,

Thank you very much for submitting your manuscript "Vibrio cholerae RbmB is an α-1,4-polysaccharide lyase with biofilm-dispersal activity against Vibrio Polysaccharide (VPS)" for consideration at PLOS Pathogens. Your manuscript was reviewed by members of the editorial board and by three independent reviewers. In light of the review from Reviewer #2, we would like to invite the resubmission of a substantially revised version that takes into account the reviewers' comments. 

Please take special care to address concerns of Reviewer #2 related to "dispersal" versus "disassembly" or "disaggregation" as well as the role of nutrient depletion in the experimental setup.

We cannot make any decision about publication until we have seen the revised manuscript and your response to the reviewers' comments. Your revised manuscript is also likely to be sent to reviewers for further evaluation.

Sincerely,

Jon Paczkowski

Academic Editor

PLOS Pathogens

D. Scott Samuels

Section Editor

PLOS Pathogens

Michael Malim

Editor-in-Chief

PLOS Pathogens

orcid.org/0000-0002-7699-2064

Reviewer's Responses to Questions

**Part I - Summary**

Reviewer #1: I am satisfied with the revisions made by the authors.

Reviewer #2: The revised manuscript by Weerasekera et al., “Vibrio cholerae RbmB is an α-1,4-polysaccharide lyase with biofilm-dispersal activity against Vibrio Polysaccharide (VPS)” focuses on the characterization of RbmB predicted to be a hydrolase and contribute to V. cholerae biofilm dispersion. Using a combination of biochemical and analytical tools, the authors demonstrated that RbmB acts as a polysaccharide lyase by cleaving VPS at a unique site into a mixture of tetramers and octamers. Tis reviewer enjoyed the enzymology and biochemical characterization of RbmB and the fate of Vps. The revisions have improved the clarity of the manuscript regarding the biochemical characterization of RbmB and the VPS analysis. And while most of the prior reviewers concerns were addressed, some concerns remain and/or the revised manuscript thanks to its improved clarity, raised a few additional questions. The main weakness for this reviewer is the contribution of RbmB to the dispersion of V. cholerae biofilms. Comments and concerns are discussed below.

Reviewer #3: I thank the authors for their thorough responses to my questions, and those of the referees. I think this is a very interesting manuscript and strongly support its publication at this time.

**Part II – Major Issues: Key Experiments Required for Acceptance**

Reviewer #1: None

Reviewer #2: 1. Previous reviewers commented on RbmB not having been shown to induce dispersion by V. cholerae biofilms in vivo and this reviewer found the response unsatisfactory. Lack of dispersion as seen for an ΔrbmB mutant does not mean that RbmB induces dispersion. There are many reasons why a biofilm may or may not disperse. And the findings reported here do not support RbmB inducing dispersion, but at most instead disassembling or disaggregation biofilms.

2. The authors state “in several places that exogenously added recombinantly produced RbmB can efficiently disperse V. cholerae biofilms and that this findings provides direct support for the ability of RbmB to induce dispersion. See also title. However, this is not considered to be dispersion but disassembly or disaggregation, as it is not an active process, but passive due to assumed matrix degradation. This reviewer highly recommends replacing “disperse” with “disassemble” or “disaggregate”.

3. While the author demonstrate that exogenously added RbmB induces biofilm disaggregation, the authors indicated that this response was only observed upon depletion of nutrients. This finding raises several questions considering that nutrient depletion alone induce dispersion by the WT . 1) why does exposure to exogenously added RbmB not induce dispersion without the added dispersion cue of nutrient depletion? What is the appearance of rbmB mutant biofilms post 1 hour of nutrient depletion? What is the effect of heat inactivated RbmB in the absence and presence of nutrient depletion, following the same incubation time?

4. An additional answered question is whether incubation of WT biofilms with recombinantly produced RbmB indeed disassembly biofilms by degrading VPS? This reviewer strongly suggests assessing the fate of VPS in vivo upon exposure to recombinantly produced RbmB.

5. Similarly, the authors should consider treating pre-formed wild-type biofilms (not rbmB mutant) with different concentrations of purified RbmB and/or inactive RbmB and analyzing the biofilm disassembly rate. This might be a direct way of showing RbmB function, instead inducing dispersion using depletion of nutrient.

6. Figure 6C - Previous reviewers commented on fluorescent images shown in Figure 6C to appear to be of different zoom which the authors assured was not the case. But what is the explanation of clusters shown in the top panel to appear larger than those in the panels below? The images represent biofilms formed by the same strain. Please explain.

7. For all graphs, please show results (data points) of individual repeats instead of just the average/stdev

8. Figure legends and stats – indicate in the legend what was used as comparator for the statistical analysis and which statistical analysis was performed (e.g. Anova will be followed by t-tests for pair-wise comparisons with Tukey’s correction?

9. IT is unclear if experiments shown in Figure 6 were carried out using cleaved or uncleaved DsbA-RbmB. The information is neither provided in the Material section of the figure legend. Other experiments/methods provide this information.

Reviewer #3: None.

**Part III – Minor Issues: Editorial and Data Presentation Modifications**

Reviewer #1: None

Reviewer #2: NA

Reviewer #3: None.

PLOS authors have the option to publish the peer review history of their article (what does this mean?). If published, this will include your full peer review and any attached files.

Reviewer #1: No

Reviewer #2: No

Reviewer #3: No
---

## [Editor Report · Decision Letter 2]

13 Nov 2024

Dear Dr. Olson,

We are pleased to inform you that your manuscript 'Vibrio cholerae RbmB is an α-1,4-polysaccharide lyase with biofilm-disrupting activity against Vibrio Polysaccharide (VPS)' has been provisionally accepted for publication in PLOS Pathogens.

Best regards,

Jon Paczkowski

Academic Editor

PLOS Pathogens

D. Scott Samuels

Section Editor

PLOS Pathogens

Michael Malim

Editor-in-Chief

PLOS Pathogens

orcid.org/0000-0002-7699-2064
---

## [Editor Report · Acceptance letter]

20 Nov 2024

Dear Dr. Olson,

We are delighted to inform you that your manuscript, "*Vibrio cholerae* RbmB is an α-1,4-polysaccharide lyase with biofilm-disrupting activity against Vibrio Polysaccharide (VPS)," has been formally accepted for publication in PLOS Pathogens.

Best regards,

Michael Malim

Editor-in-Chief

PLOS Pathogens

orcid.org/0000-0002-7699-2064